# Effects of Supports BET Surface Areas on Membrane Electrode Assembly Performance at High Current Loads

Paritosh Kumar Mohanta *, Masuma Sultana Ripa, Fabian Regnet and Ludwig Jörissen

Zentrum für Sonnenenergie-und Wasserstoff-Forschung Baden-Württemberg, Helmholtzstrsse 8, 89081 Ulm, Germany; masuma-sultana.ripa@zsw-bw.de (M.S.R.); fabian.regnet@zsw-bw.de (F.R.); ludwig.joerissen@zsw-bw.de (L.J.)
* Correspondence: paritosh.mohanta@zsw-bw.de; Tel.: +49-(0)731-95-30-211

**Abstract:** In this work, we investigated the influence of catalyst supports, particularly Brunauer, Emmett, and Teller (BET) surface area of the catalyst support materials, on membrane electrode assembly (MEA) performance. Keeping the anode catalyst layer (CL), membrane, Pt loading, and operating condition unchanged, we prepared cathode CLs using catalysts of identical Pt content (30 wt% Pt) which were supported on carbon black materials having different BET surface areas. We observed optimum cell voltage at high current load when using cathode catalyst layers prepared from catalysts supported on carbons having medium-BET surface area. High-BET surface area supports, although beneficial at low current density as well as low-BET surface area supports, led to increased voltage losses at high current load due to mass transport limitations which can be explained by the electrochemically active surface area available and water management in the catalyst layer.

**Keywords:** PEMFC; MEA; catalyst support; catalysts; Pt; Stable carbon; RDE; ORR





## 1. Introduction

The use of polymer electrolyte membrane fuel cell (PEMFC) technology is increasing steadily due to its high-energy conversion efficiency at zero emission. However, the major hurdles for commercialization are the still high noble metal demand and components' durability in real-world application. To reduce cost, high surface area platinum nanoparticles are often supported on high Brunauer, Emmett, and Teller (BET) surface area electrically conductive support materials such as carbon black (CB). However, CB materials are thermodynamically unstable at the high electrode potential at low load or open circuit potential (OCP) under realistic fuel cell operating temperatures and pressures. In particular, CB in the cathode is severely oxidized during so-called air–air starts of the fuel cell, which leads to carbon corrosion causing detachment of Pt nanoparticles and promoting agglomeration of Pt nanoparticles, and eventually, performance degradation [1,2]. Nevertheless, CB is widely used as noble metal support due to its high electronic conductivities, beneficial metal supports interactions, and low costs.

In order to increase the durability of fuel cell catalyst support materials, highly graphitized CB materials were suggested in references [3–6]. However, graphitized CB materials often possess very low-BET surface areas on which preparation of highly Pt-loaded catalyst with small Pt particle size is not feasible because of agglomeration of Pt nanoparticles [7], simultaneously decreasing the Pt electrochemical active surface area (ECSA) and, thus, Pt utilization.

The introduction of surface functional groups on CB materials to increase both the stability and the oxygen reduction reaction (ORR) activities of the catalysts recently became very popular [8–13]. Improvement of sluggish kinetics of the ORR by increasing the intrinsic activities of the catalysts is likely resulting in a further reduction of the Pt demand. Graphitic hollow carbon nanocages [8], doped graphite nanofibers [9], nitrogen-doped carbons [10], and nitrogen/metal co-doped graphene tubes [11] are reported as viable

catalyst support materials for fuel cell applications. Activated carbon composites [12], xerogel–nanofiber carbon composites [13], carbon nanofibers [14,15], and silica-coated carbon nanotubes [16] have also found durable noble metal support materials for use in PEMFC. Unfortunately, most of the reported highly ORR active stable catalysts were either not transferable to membrane electrode assembly (MEA) or did not meet the expectation in real-time fuel cell operation. In our previous work, CB materials prepared by a silica template process showed promising properties as support materials in terms of both stabilities and activities for fuel cell applications [17]. However, the preparation of a highly Pt-nanoparticle-loaded catalyst is still challenging due to the low-BET surface areas of those support materials.

Inorganic oxide materials like graphitized CB are also considered durable under fuel cell operating conditions [18–21]; however, due to low electrical conductivities, they are leading to voltage losses at high current load. Doped metal oxides, for example, antimony-doped tin oxide [22–26], Nb-doped $TiO_2$ [27], indium tin oxide [28], and carbon-doped $TiO_2$ [29,30] showed significant increases of electrical conductivities compared to the bare materials; nevertheless, their electrical conductivities and BET surface areas are not high enough to be applicable for PEMFCs. Carbides [31,32] and borides [33] have also been proposed as stable cathode catalyst supports for PEMFC applications. Unfortunately, most catalysts based on the proposed non-carbon-based support materials are not transferable to the real fuel cell applications. Thus, CB is still being used as noble metal support for PEMFC.

An increase of fuel cell power density is one essential route to cost reduction. Thus, the fuel cell should be capable of operation at high current loads. Sufficient supply of reactant, the removal of product water from the reaction site, and high electronic and ionic conductivity are crucial to sustaining a high reaction rate at the catalytic sites. Too low pore volume or flooding the porous catalyst layer, for example, would induce significant voltage losses due to mass transport effects. Therefore, the properties of catalyst support materials, as well as catalyst layer (CL) morphologies, play significant roles in MEA performance.

In this work, we studied MEAs under high current load ($>1$ A cm$^{-2}$) using cathode CLs made from catalysts supported on CB of different BET surface areas while keeping the anode CL and the electrolyte membrane constant. Both homemade and commercial catalysts of 30 wt% Pt content were chosen for this investigation. Homemade catalysts were prepared using a modified polyol method and stabilized by annealing under reducing atmosphere [34–37]. All catalyst materials used in MEA studies were thoroughly pre-characterized with respect to stability and activity.

During air–air start-up of a fuel cell, the electrode potential is raised to extremely high potentials ($>1.1$ V) resulting in the oxidation of carbon support materials [1,38–40]. Thus, a simulated start–stop cycle was used to assess the stability of the support materials herein.

Using a rotating disk electrode (RDE) is a very common and simple method to investigate the ORR activities of the catalysts; however, the outcomes are very sensitive to the purity of the chemicals, ink preparation techniques, catalyst thin-film quality on the RDE, voltage scan rate, and appropriate data corrections [41–44]. Nonetheless, the activities of the catalysts were measured via RDE in this work.

Finally, both homemade and the reference catalysts were processed to MEAs for comparative single cell performance testing via measuring the so-called polarization curves. Electrochemical impedance spectroscopy (EIS) which is a common technique for MEA investigation was also carried out. At the end of all MEA tests, morphologies of the CLs were investigated via Focussed Ion Beam Scanning Electron Microscopes (FIB-SEM) images.

## 2. Results

### 2.1. Physical Characterization of the Supports and the Catalysts

Three different types of support materials—a carbon black (Vulcan XC72, Cabot Corporation, Billerica, MA, USA), a Timcal super C 65 (C65) (Imerys, France), and a Timcal

graphitized carbon RE167 (Imerys, Paris, France)—were chosen for variation of the catalyst support materials' BET surface areas in this work.

Both Timcal super C 65 and RE167 were hydrophobic in nature which is unfavorable to deposit Pt via the polyol synthesis process. Thus, they were pre-oxidized in air at elevated temperature to introduce oxidized surface groups to improve their wettability. The change of hydrophobic properties of Timcal graphitized carbon RE167 (GC) after oxidation was observed by visual investigation by adding a drop of water to the powder samples, as described elsewhere [6]. In this work, contact angles of the oxidized C65 were additionally measured. Bare super C65 support was very hydrophobic and showed low-BET surface area (62 $m^2$ $g^{-1}$). After oxidation in air, the contact angle of the support material decreased from 31° to 19.6°, which is very close to the contact angle of the Vulcan XC72 CB material (15.1°). The hydrophobic property was also found to be decreased by visual observation of the wetting behavior. Additionally, oxidation of the super C 65 support increased the BET surface area from 62 to 174 $m^2$ $g^{-1}$. The change of surface properties of the support materials was due to introduction of some surface functional groups to the carbon support that was previously confirmed via Böhm titration [17]. Nevertheless, the BET surface areas, electronic conductivity, and hydrophilicity of oxidized super C65 were still lower than the traditional Vulcan XC72 (CB) material as shown in Table 1.

**Table 1.** Physical characteristics of the chosen support materials.

| Supports | Supports BET, $m^2$ $g^{-1}$ | Supports El. Conductivity, S $cm^{-1}$ |
|---|---|---|
| Vulcan XC72 (CB) | 192 (bare) | 2.3 |
| Super C65 (C65) | 174 (oxidized) | 1.0 |
| Timcal RE167 (GC) | 63 (oxidized) | 2.1 |

BET = Brunauer, Emmett, and Teller, CB = carbon black, C65 = Timcal super C 65, GC = Timcal graphitized carbon RE167.

On as-received CB and oxidized supports, 30 wt% Pt-containing catalysts were prepared by using an already developed modified polyol process. Two commercially available catalysts of the same Pt content (30 wt%) supported on medium- and high-BET surface area supports were chosen as benchmarks for this investigation.

In fuel cells, uniformly dispersed Pt nanoparticles on support materials are highly desired in order to increase the ORR kinetics as well as Pt utilization. Homogeneous Pt-particle distribution of the homemade catalysts by comparison of X-ray Powder Diffraction (XRD) data and Transmission Electron Microscope (TEM) measurements was already shown elsewhere [45].

Table 2 shows that the particle sizes of the homemade catalysts were in the range of 3 to 7 nm, thus, meeting our targets, in terms of the Pt content and the Pt crystallite sizes, although their average Pt crystallite sizes are still larger than the ones observed in commercial catalysts.

**Table 2.** Physical characteristics of both homemade and commercial catalysts.

| Catalyst | Company | Supports | Supports BET, $m^2$ $g^{-1}$ | Actual Pt, wt% | Pt Crys., nm |
|---|---|---|---|---|---|
| TKE | Tanaka | High surface area CB | 800 * | 28.2 * | 0.7 |
| TKV | Tanaka | Vulcan XC72 | 250 * | 28.9 * | 1.4 |
| Pt/CB | Homemade | Vulcan XC72 | 192 | 28.2 | 2.9 |
| Pt/C65 | Homemade | Super C65 | 174 | 29.1 | 3.3 |
| Pt/GC | Homemade | Timcal RE167 | 63 | 28.7 | 6.8 |

* taken from data sheet supplied by the manufacturer.

Both CB- and C65-supported catalysts showed almost the same average Pt particle sizes since they have almost the same BET surface areas of the support materials. However, due to the low-BET surface area of GC, larger Pt nanoparticles were formed. Pt nanoparticles deposited in close proximity have a high tendency to minimize their surface

energy, eventually forming larger average Pt particles on the low-BET-surface GC support compared to the CB and C65 supports where larger interparticle distances are possible.

XRD patterns of the catalysts are shown in Figure 1. The peak broadenings of both commercial catalysts are evidence of the smaller Pt crystallite sizes compared to the home-made catalysts.

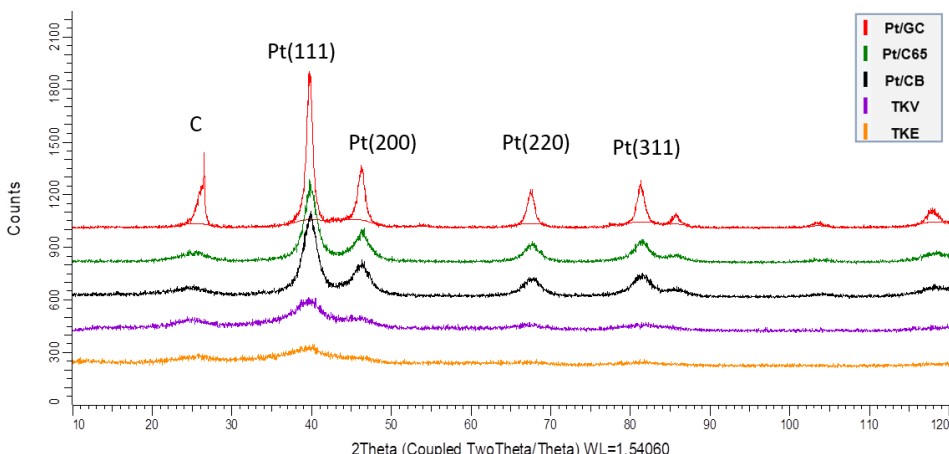

**Figure 1.** XRD patterns of the catalysts.

### 2.2. Supports Corrosion

Traditional CB catalyst support materials are severely corroded during start-up and shutdown of a fuel cell leading to Pt detachment, agglomeration, and, ultimately, low performance of the fuel cell. It has been shown in previous work [17] that the rate of supports corrosion in a three-electrode setup in a liquid electrolyte and an in situ MEA test are identical. Thus, we simulated the effects of start–stop cycles in a three-electrode setup in this work to assess carbon corrosion. Figure 2 shows the change of ECSA survival rates of the supported catalysts while the working electrode potential was cycled between 1.0 and 1.5 V vs. reversible hydrogen electrode (RHE) at a 0.5 V s$^{-1}$ voltage scan rate. After the end of the tests (60,000 cycles), the highest ECSA survival rate (final ECSA/initial ECSA in percentage) was found from the catalysts prepared with GC support followed by C65 and CB supports. Apart from the GC and C65, all catalysts supported on CB materials showed almost the same corrosion rate. Therefore, we expect improved long-term fuel cell operation from the catalyst supported on the GC and C65 supports compared to the traditional CB supports. In our previous work [6], the ECSA survival rate of 20 wt% Pt on GC was found to be 92%; however, we observed no net carbon corrosion in this work with the same support material. It could be due to less available carbon surface (since 28.7 wt% Pt on GC) and/or larger Pt particle sizes compared to the previous work.

While plotting cyclic voltammograms (CVs) of all catalysts in between the tests, a clear view of a change of CV patterns can also be evidence of the better stability of GC and C65 compared to other CB support materials (see Figure S1).

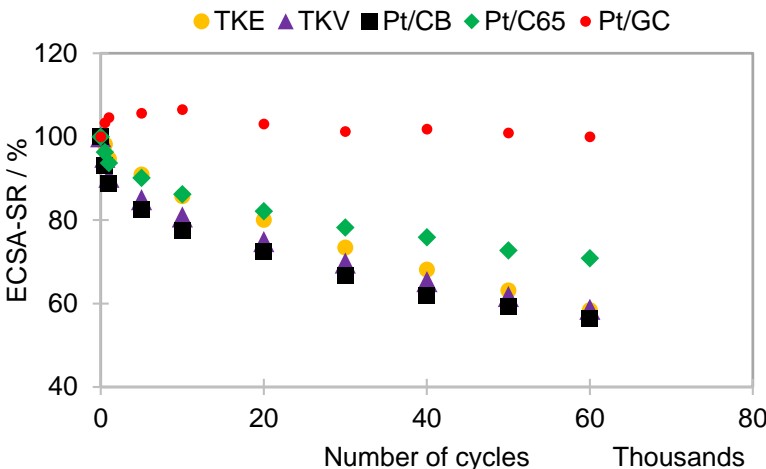

**Figure 2.** Changed electrochemical active surface area (ECSA) survival rates of the supported catalysts while potential cycling between 1.0 and 1.5 V vs. RHE at 0.5 V s$^{-1}$ voltage scan rate at room temperature (RT).

### 2.3. ECSA and ORR Activities of the Catalysts

Figure 3 shows CV patterns (in the RDE setup) of the catalysts in which hydrogen adsorption–desorption currents are likely dependent on the crystallite sizes of the Pt nanoparticles. However, double-layer (DL) capacitive currents are possibly depending on the BET surface areas and the surface functional groups of the support materials. As expected, catalysts with the lowest (TKE) and the highest (Pt/GC) Pt crystallite sizes are showing the highest and the lowest hydrogen adsorption–desorption currents, respectively. Likewise, the catalysts supported on the highest BET surface areas (TKE) showed the highest DL capacitive currents.

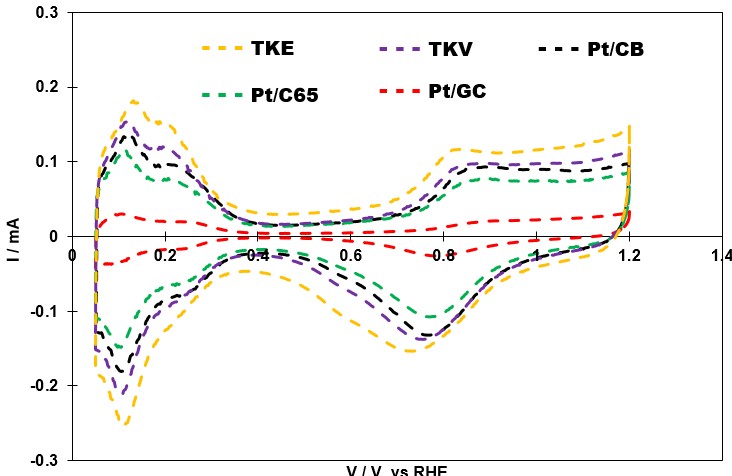

**Figure 3.** Cyclic voltammograms (CVs) of the supported catalysts at 50 mV s$^{-1}$ in 0.1 M HClO$_4$ at RT, 2 μg Pt on rotating disk electrode (RDE).

A highly active catalyst can reduce the sluggish kinetics of the ORR reaction and ultimately increase the performance of the fuel cell. Thus, measurements of mass activities (MAs) at 0.9 V vs. RHE and ECSA of the catalysts in 0.1 M HClO$_4$ were performed (see Figure 4). The highest ECSA of the TKE, due to the lowest average crystallite sizes of Pt, among others, was expected. Likewise, ECSAs were obtained from TKV, Pt/CB, Pt/C65, and Pt/GC catalysts. The lower mass activities of the homemade catalysts were due to low Pt crystallite sizes of the catalysts compared to the commercial catalysts.

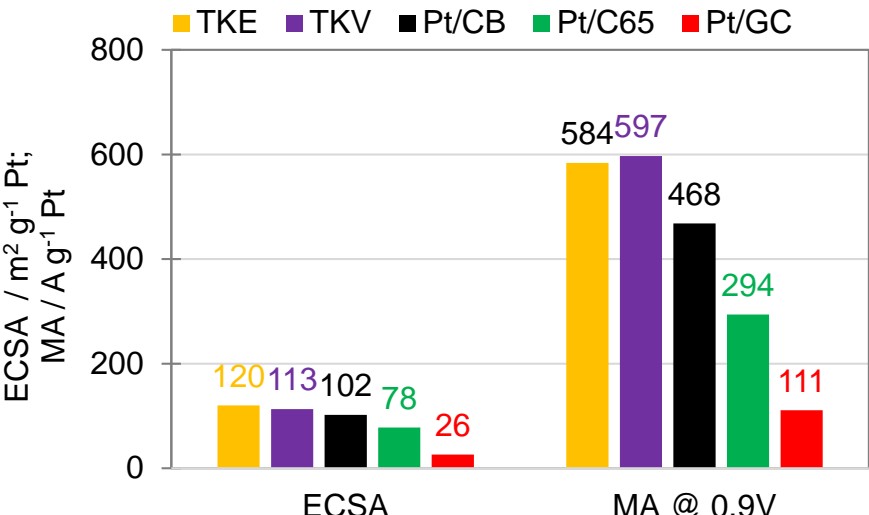

**Figure 4.** Comparison of mass activities (at 0.9 V vs. RHE) and ECSA of the catalysts at 50 mV s$^{-1}$ voltage scan rate in 0.1 M HClO$_4$ at RT.

## 2.4. MEA Performance and Diagnostics

In order to compare the MEA performance of the supported catalysts, MEAs should have been prepared and tested in a similar manner. However, the amount of ionomer in the catalyst layers (CLs) can change the fuel cell performance at high current densities dramatically. A thick layer of ionomer film in the CL increases the layers' protonic conductivity while increasing the reactant transport resistance. On the contrary, the permeability of oxygen is found to increase with a decrease of ionomer film in the CLs [46]. Therefore, the ionomer content in the CL plays a crucial role in optimizing MEA performance by simultaneously minimizing losses caused by mass transport as well as electronic and proton conductivity [47–49]. In previous work, we found the optimum ionomer to carbon ratio (I/C) is to be dependent on the Pt content of the catalysts and the BET surface areas of the support materials [6,17,50]. Primarily, we found the optimum I/C ratio to be proportional to the BET surface areas of the support materials [6,17]. Yet, when using the same support material, the optimum ionomer amount was inversely proportional to the catalysts' Pt content [50]. Unfortunately, there are no established tools available to define the optimum I/C ratio purely from the material parameters. Thus, we prepared MEAs with varied I/C for each catalyst and subsequently selected the optimum composition on the basis of the I–V performance at high current density (see Figure S2).

Figure 5 displays I–V characteristics of the MEAs with optimized I/C (see Table 3) in the cathode CLs using the same noble metal loading and operating conditions. At low current loads (<1 A cm$^{-2}$), MEAs prepared with the catalysts TKE and TKV showed higher voltages, compared to the MEAs prepared with the homemade catalysts. The intrinsic catalytic activities were likely the dominating factor, i.e., catalysts with low Pt crystallite sizes (high ECSA) showed high voltages. However, at high current loads (>1 A cm$^{-2}$), a homemade catalyst (Pt/C65) shows the highest voltages, although its ORR activity (MA) was lower than the reference catalysts. Apparently, at high current densities, the CL morphologies and water management are the dominating factors rather than the intrinsic activities of the catalysts.

Although TKE showed the highest voltages at low current loads, due to the largest BET surface areas of the support materials, smaller Pt particles were possibly hiding inside the porous structure in which ionomers were not accessible, resulting in a reduction of the Pt utilization at high current loads [51]. On the other hand, improved voltages at high current loads were achieved from the MEAs prepared with the medium (TKV, Pt/CB, Pt/C65) surface areas supported catalysts, assuming no or fewer Pt nanoparticles are hiding in non-accessible pores resulting in better Pt utilization. A clear image of the I–V characteristics at defined current loads is shown in Table 4 (extracted from Figure 5).

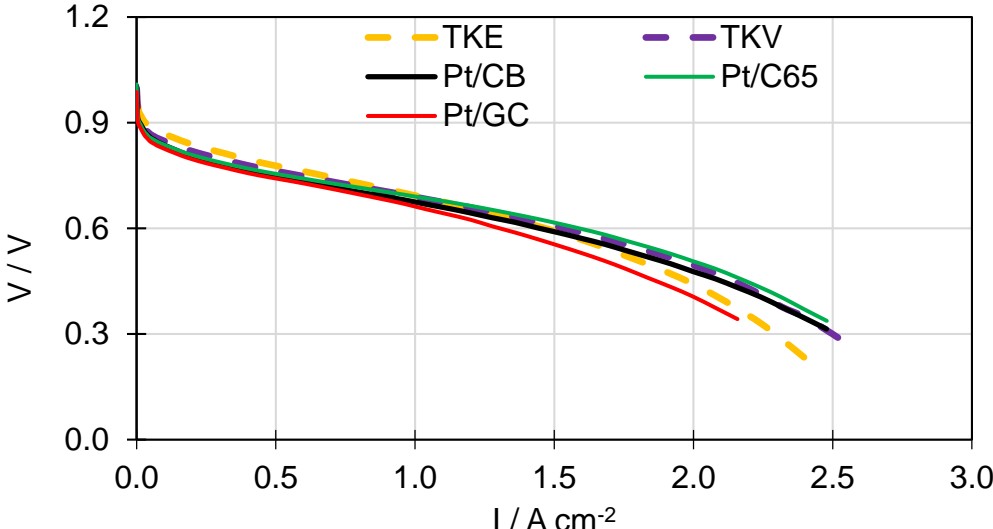

**Figure 5.** I–V characteristics of the membrane electrode assemblies (MEAs) at 150 kPA under cell operating temperature 80 °C, anode and cathode dew point 80 °C, inlets 84 °C, and anode and cathode stoichiometry 1.3 and 3, respectively.

**Table 3.** Optimum ionomer to carbon ratio (I/C).

| Catalyst | Supports | Supports BET $m^2 \, g^{-1}$ | Optimum I/C |
|---|---|---|---|
| TKE | HSCB | 800 | 0.74 |
| TKV | Vulcan XC72 | 250 | 0.72 |
| Pt/CB | Vulcan XC72 | 192 | 0.72 |
| Pt/C65 | Timcal Super C65 | 174 | 0.82 |
| Pt/GC | RE167 | 62 | 0.5 |

**Table 4.** Voltages (in V) of the MEAs at defined load currents.

| Catalyst | OCP | 0.04 A $cm^{-2}$ | 0.16 A $cm^{-2}$ | 1.36 A $cm^{-2}$ | 2.16 A $cm^{-2}$ |
|---|---|---|---|---|---|
| TKE | 0.999 | 0.896 | 0.849 | 0.624 | 0.372 |
| TKV | 0.997 | 0.875 | 0.829 | 0.632 | 0.445 |
| Pt/CB | 1.004 | 0.867 | 0.816 | 0.617 | 0.432 |
| Pt/C65 | 1.009 | 0.867 | 0.818 | 0.640 | 0.460 |
| Pt/GC | 0.987 | 0.856 | 0.806 | 0.588 | 0.342 |

Electrochemical impedance spectroscopy can distinguish between different electrochenmical and mass transport processes. The so-called Nyquist plot is often used for graphical analysis of the impact of different impedance elements in a fuel cell during operation. Figure 6 shows the Nyquist plots of the MEAs operated at 0.16 and 2.16 A $cm^{-2}$ current loads. At 0.16 A $cm^{-2}$, the sequence of the low-frequency arc diameter of the TKE followed by TKV, Pt/C65, Pt/CB, and Pt/GC are absolutely following the I–V characteristics curve (also see Table 4). However, at 2.16 A $cm^{-2}$, Pt/C65 shows the lowest low-frequency-arc diameter followed by Pt/CB, TKV, Pt/GC, and TKE, indicating high reactant transportation perhaps due to maximum Pt utilization and proper water management of the Pt/C65 among others which were also reflecting at the I–V characteristics curves of the MEAs.

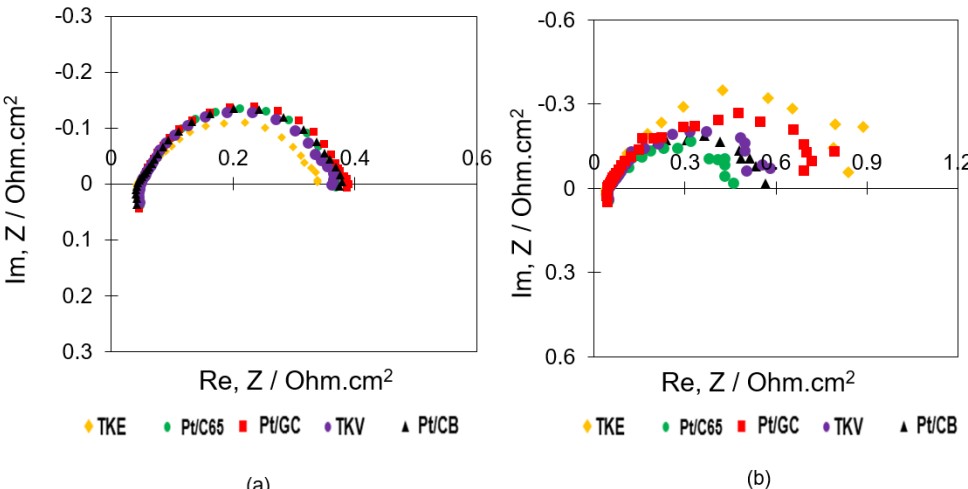

(a)　　　　　　　　　　　　　　　　　　　(b)

**Figure 6.** Nyquist plots of the MEAs at (**a**) 0.16 and (**b**) 2.16 A cm$^{-2}$ current loads.

At high current density besides the intrinsic catalytic activity, reactant transport and water management were becoming decisive for the cathode performance. The rather large noise in the low-frequency part of the impedance spectrum indicated the onset of electrode flooding which was most prominent in the case of TKE and Pt/GC.

A thick CL can increase ohmic and reactant transport resistance of the MEA simultaneously; it can also impede the removal of product water. Both can adversely affect MEA performance [50]. We assumed, together with the high ohmic resistance, the low Pt utilization of TKE was likely caused by a rather thick CL compared to the other catalysts since the FIB-SEM images reveal the average CL thickness of TKE, TKV, Pt/CB, Pt/C65, and Pt/GC were 19, 17, 12, 16, and 15 μm, respectively (see Figure S3). Likewise, the morphology of the cathode CL could be involved in proper water management of the fuel cell, which can affect the reactant flow to the reaction sites. A different cathode CL morphology of the TKE was observed in the FIB-SEM images that could also be unfavorable at high current loads assuming flooding compared to the others (see Figure 7).

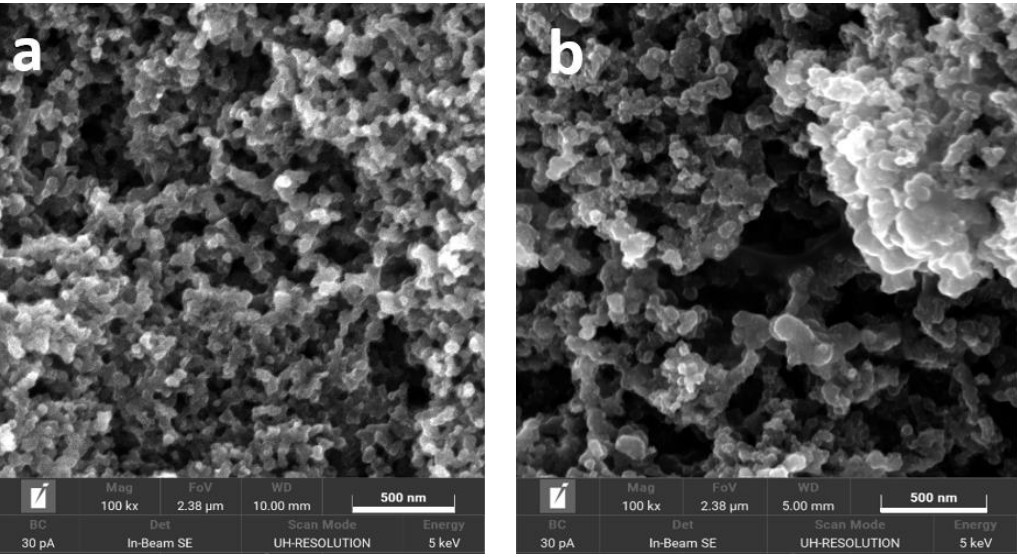

**Figure 7.** *Cont.*

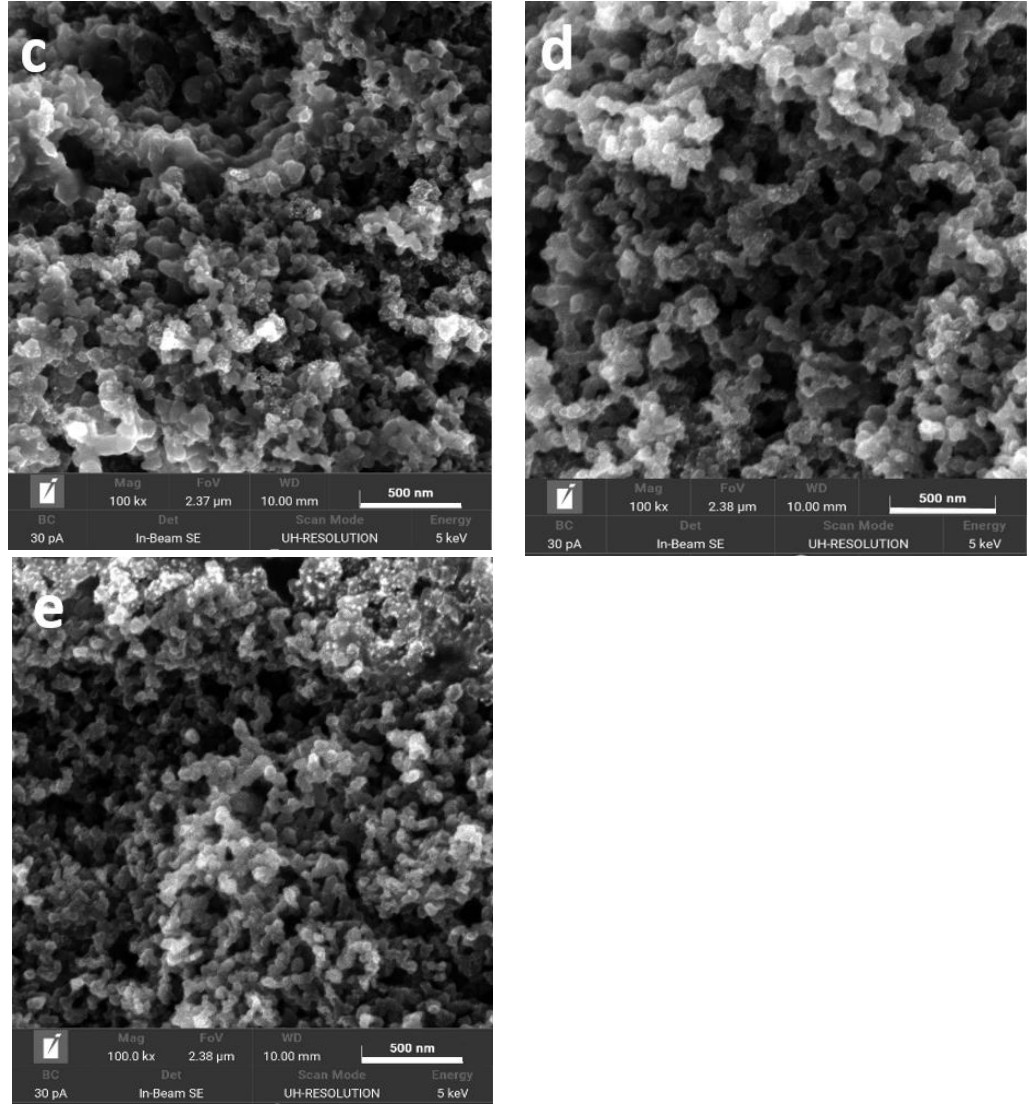

**Figure 7.** FIB-SEM images of the cathode CLs (cross-section) of the MEAs prepared with (**a**) TKE, (**b**) TKV, (**c**) Pt/CB, (**d**) Pt/C65, (**e**) Pt/GC catalysts.

### 3. Summary and Conclusions

The performance of a PEMFC is dependent on several components such as catalysts, catalyst supports, membranes, CL compositions, gas diffusion layer, reactant flow fields, and operating conditions. Among others, the selection of the catalyst support materials can play a vital role, as it can change the CL morphology and water management, subsequently, the durability and the MEA performance. Thus, the proper design of a CL is crucial to increase the power density since it can increase Pt and reactant utilization by improving reactant transport and water management at high current loads. Typically, high-BET surface area support materials are used to prepare fuel cell catalysts. However, BET surface areas of the catalyst support materials play an important role while the fuel cell is subjected to operate at high current loads. In this work, the designing of cathode CLs was done with identical Pt (30 wt%)-containing catalysts but different supports' BET surface areas to investigate the MEA performance at high current loads (>1 A cm$^{-2}$).

On supports materials having different BET surface areas (between 60 and 200 m$^{-2}$ g$^{-1}$), 30 wt% Pt-containing electrocatalysts were successfully prepared. Two commercially available catalysts supported on medium (250 m$^{-2}$ g$^{-1}$) and high (800 m$^{-2}$ g$^{-1}$) BET surface

areas materials with the same Pt content (30 wt%) as homemade catalysts were investigated as benchmarks.

In order to assess the durability of the support materials, catalysts were subjected to simulated start–stop cycle tests. Homemade catalysts supported on low-BET surface areas were found to be more durable compared to all other CB-supported catalysts in terms of ECSA survival rate after 60,000 potential cycles between 1.0 and 1.5 V vs. RHE in 0.5 M $H_2SO_4$ at room temperature (RT). However, the ORR activity (at 0.9 V vs. RHE in RDE set up) of the catalysts was found to be lower due to larger average Pt crystallite sizes compared to the reference catalysts.

As expected, MEAs prepared with the catalyst supported on low-BET surface area (62 $m^2$ $g^{-1}$) graphitized carbon showed the lowest voltage under low- and high-load conditions which can be explained by the lowest ECSA of the catalyst materials studied in this work. Furthermore, the large coverage of the support by Pt metal likely enhanced the tendency for catalyst layer flooding.

MEAs prepared with the TKE catalyst (800 $m^2$ $g^{-1}$) showed the highest cell voltages at low current loads (<1 A $cm^{-2}$) which can be explained by the highest ECSA of the catalysts studied in this work. The rather high mass transport losses at high current densities were likely due to water accumulation in small pores, thus, preventing reactant access to the Pt nanoparticles deposited in there.

Surprisingly, MEAs prepared with catalysts supported on medium-BET surface areas (174–250 $m^2$ $g^{-1}$) despite showing comparatively low voltage at low current densities, showed high voltage at high current loads indicating that the overall MEA performance not only depends on the intrinsic catalytic activities but also on the CL compositions and morphologies. In this case, an optimum I/C ratio and catalyst support structure can ensure optimum Pt utilization (i.e., no idle Pt inside the pore structure even at high current load), as well as reactant supply (avoidance of mass transport effects by sufficient open pore volume by proper water management).

In conclusion, we find cathode catalysts supported on medium-BET surface area supports advantageous to the traditionally used high-BET area support materials when fuel cell operation under high current load is concerned. Graphitized support materials, despite having high corrosion stability, are less suitable for use in a cathode catalyst layer since Pt nanoparticles show a tendency to agglomerate on these low surface area supports reducing ORR activity and promoting the tendency to catalyst layer flooding.

Finding medium-BET surface area support material with improved corrosion resistance and high electronic conductivity remains a big challenge to achieve high-performance durable MEA for fuel cell applications.

## 4. Materials and Methods

A Vulcan XC72 (CB), a Timcal super C 65 (C65), and a Timcal RE167 (GC) catalyst support materials were chosen for investigation. C65 material was oxidized at 550 °C in an air oven for 1 h in order to improve the hydrophilicity of carbon materials by introducing oxygen-containing surface groups [52,53]. After oxidation, the wetting behavior of the materials was investigated by measuring the contact angle. The measurement was done by spraying the carbon ink onto a sheet of Polytetrafluoroethylene (PTFE) followed by drying the change of hydrophobicity, evaluated using a drop shape analyzer (Krüss, Hamburg, Germany). As received, the GC was also hydrophobic in nature. Thus, it was oxidized at 600 °C for 90 min in an air oven [6]. The CB material was used without any further treatment.

The BET surface areas of the materials were measured by using a Sorptomatic 1990 instrument (Thermo Scientific Inc, Waltham, MA, USA), while nitrogen was used as an adsorbent.

A highly conductive catalyst support material is desirable to reduce the ohmic resistance of the CLs. Thus, the electronic conductivities of the support materials were measured by using a four points homemade device as described elsewhere [45] for comparison.

On the chosen support materials, 30 wt% Pt-containing catalysts were prepared by using an already developed modified polyol process. The synthesis procedure was identical to the process described in our previous work [50]. The prepared catalyst was annealed at 250 °C for 1 h in reducing atmosphere (95 vol% Ar and 5 vol% $H_2$) [45].

Pt contents of the prepared catalysts were determined by using an inductively coupled plasma optical emission spectrometry (ICP-OES) technique in an ICP-OES spectrometer (Acros FHS12, Spectro Analytical Instruments GmbH, Kleve, Germany). Thirty mg of each catalyst powder was digested in 2 mL of aqua regia solution (1.5 mL of 32% HCl from Sigma-Aldrich and 0.5 mL of 65% $HNO_3$ from Fluka) overnight to dissolve the Pt completely. The mixture was then diluted to 25 mL with ultrapure water. After filtration, the clear filtrate was used to measure the Pt content via ICP-OES spectrometer. A Siemens D5000 XRD instrument was used to determine the crystallite size of Pt by using TOPAS software (Bruker AXS, Version 5).

A triangular wave potential cycle test in a Zahner IM6 potentiostat (Zahner-elektrik GmbH & Co. KG, Kronach, Germany) was performed to accelerate supports' corrosion in a three-electrode setup at room temperature [54]. A Pt wire and a $Hg/Hg_2SO_4$ were used as counter and reference electrodes respectively. The working electrode (WE) was prepared by placing 8 µg Pt-equivalent inks on a glassy carbon disk, which was then dried in open air. The ink was prepared by stirring 10 mg of catalyst powder in 5 mL of solvent (0.05 wt% Nafion® in 10 vol% isopropanol in water) for 2 min followed by sonicating in an ultrasonic bath for 15 min. The mixture was additionally stirred for 2 min before transferring the ink with a micropipette (Corning Lambda plus) to the glassy carbon disk. Then, 0.5 M $H_2SO_4$ solution was taken in an electrochemical cell. Next, nitrogen gas was bubbled through the electrolyte for 10 min to make the electrolyte oxygen free. The WE was then immersed in the electrolyte and was cleaned electrochemically by recording CVs between 0.05 and 1.2 V vs. RHE at a 150 mV s$^{-1}$ scan rate until a reproducible cyclic voltammogram (CV) was reached. Subsequently, five CVs were recorded at 50 mV s$^{-1}$ voltage scan rate to measure the initial ECSAs of the catalysts. The ECSAs were calculated from the average value of hydrogen adsorption ($Q_{Hads}$) and desorption ($Q_{Hdes}$) charges in the potential range between 0.05 and 0.4 V vs. RHE from the voltammogram as described in reference [54]. The start–stop cycles were carried out with a triangular wave potential cycling between 1.0 and 1.5 V vs. RHE at a sweep rate of 0.5 V s$^{-1}$. After 500, 1000, 5000, 10,000, and then every 10,000 cycles until 60,000 cycles, CVs were recorded to determine the ECSA by the same ways mentioned above.

ORR activities of the catalysts were measured in a three-electrode setup at room temperature using a rotating disk electrode (RDE) as described elsewhere [50]. Mass activities of the catalysts at 0.9 V vs. RHE were then calculated from the measured data using a Koutecky-Levich plot [45,54].

Catalyst-coated membranes (CCMs) of 25 cm$^2$ active surface areas were prepared by a manual spray-coating technique using an airbrush. The ink for the electrodes was prepared by stirring the required amount of catalysts, ultrapure water, and Nafion® solution (20 wt%, D2021 Ion power) for 4 h. A Gore select (15 µm) membrane was chosen to prepare all MEAs. Gore membrane has two specific sides; coated blackish side was always used as cathode throughout this work (recommended by the manufacturer). Anode and cathode Pt loadings were set to 0.1 and 0.25 mg cm$^{-2}$ respectively. The anode CL was prepared with a Tanaka catalyst (TEC10V30E, 28.3 wt% Pt), in which a 0.72 ionomer to carbon (I/C) ratio was used. However, the amount of ionomer on the cathode CLs was chosen at an optimum level, i.e., where the highest MEA performance at high current density was achieved [17].

CCMs were pressed at 100 bar at 140 °C for 4 min and were then sandwiched between two gas diffusion layers (SGL, BC29) in a fuel cell technology (FCT) single-cell housing that contained triple serpentine flow patterns. It was then tightened sufficiently (at 8.5 Nm torque per screw) to make the system leakage free as well as to ensure the proper contact of all components.

A Greenlight Innovation test station (G20) was used to test all MEAs under the same operation conditions. Initially, break-in of MEA was done by applying a square type load cycling between 12 and 16 A with 10 min holding times of each point in a repeating manner for five hours (see Figure S4). The polarization curve was obtained by applying the load from the maximum load to the no-load currents (descending) and from the no-load to the maximum load currents (ascending). After the test, data were evaluated by taking the average cell voltage from the last 30 s measurement data of each point using a homemade chart tools software based on Excel. I–V curve was then plotted by taking the average value of descending and ascending voltage and currents (see Figure S5).

Electrochemical impedance spectra (EIS) of the MEAs were performed to investigate the contribution of different components' impedance on the MEA performance at a defined load current. A Zahner P241 potentiostat in combination with a Greenlight Innovation test bench was used to conduct the test. The measurement was performed just after the completion of the single-cell performance test in each case. Finally, MEAs were broken by using liquid nitrogen in order to investigate the CLs' morphologies via a FIB-SEM (Tescan S9000, Dortmund, Germany) instrument.

**Supplementary Materials:** The following are available online at https://www.mdpi.com/2073-4344/11/2/195/s1, Figure S1: Change of CV patterns of the supported catalysts after 0, 10,000, 30,000, and 60,000 potential cycles; Figure S2: Optimization of I/C of Pt/C65 catalyst; Figure S3: FIB-SEM images of the cathode CLs; Figure S4: Example of an MEA conditioning and testing procedure; Figure S5: Example of a polarization curve (data evaluation).

**Author Contributions:** This work was designed by L.J. and P.K.M., P.K.M. and M.S.R. implemented the overall work under the supervision of L.J., F.R. conducted the FIB-SEM analysis. All authors have read and agreed to the published version of the manuscript.

**Funding:** This research was funded by the Federal Ministry of Transport and Digital Infrastructure (BMVI) Germany, grant number 03B10103A/1 and 03B10103A/2.

**Conflicts of Interest:** The authors declare no conflict of interest.

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
