# Peer review of "Effects of Supports BET Surface Areas on Membrane Electrode Assembly Performance at High Current Loads"

_catalysts, doi:10.3390/catal11020195_

Round 1

Reviewer 1 Report

Mohanta et al. thoroughly investigated several different catalysts and catalyst supports in PEMFCs. The team used many experimental techniques to show that having a catalyst support with low BET surface area improves performance. I commend the authors on the breadth of techniques and number of experiments carried out. Unfortunately, the experiments carried out did not seem to support their conclusions and the experiments would have been more convincing if the authors had been more systematic in looking at the effect of a single dependent variable on an independent variable while keeping everything else constant. Many things seem to be changing within the experiments, which makes it tough for the reader to understand the effect of the different variables. In addition, the quality of the figures and writing needs significant improving before publication.

Major comments

  1. You state that at high currents that supports with low BET surface area are best. However the C65 support has the best performance when comparing all systems using the homemade Pt and it has a very high BET area. The Timcal has a lower BET surface area and has worse performance. I am confused about the conclusions made. I would strongly suggest to keep the thickness of the layers the same so it is easier to compare as well.
  2. The motivation for the work is also not apparent. Much of the introduction is spent on more general topics, but there is no discussion of the work that is similar to their own which there is a lot of. I would suggest to provide a thorough literature review of studies on catalyst supports and the effect of their surface areas etc. on performance. If this has never been done than state that and what is the work that is the most similar and what does it say.

Other comments

  1. Figure 1 legend doesn’t correspond to one column in the table above it. Data on how the crystallite size was determined and the error would be useful.
  2. Table 2 what is the high surface area CB? I don’t see it in table 1.
  3. Figure 2 would be useful to explain why the one sample has particles that seem to decrease in size. Is this actually happening or is this a measurement error?
  4. Figure 3 why not normalize the current by surface area? This would be a more fair comparison. Is the quantity of current in this test really correlate with ORR kinetics? Why not measure ORR currents? Is this what you are trying to do in Figure with with MA? I am not sure what MA is, but you should show more about the KL procedure you used.
  5. How is the optimum I/C found? Is that from literature or is that something you are measuring?
  6. Figure 6 is very sloppy. The figures should be the same size. You should be showing trends in fitted resistance and capacitance in addition to the Nyguist plots.
  7. Figure 7 should have the bottom removed and to only leave the scale bar. Then label on the images in white the sample and put them together on a figure that can be on one page.
  8. Higher quality figures are expected for publication.
  9. You never discuss your supplementary figures in the main text.

Comments on writing

  1. Acronyms in the abstract are not defined.
  2. The first four paragraphs of the discussion section are not a discussion of your results.
  3. The writing is understandable, but many sentences like the following couple I show here as an example need proofreading before being published:
    1. Introduction of surface functional groups on CB materials are recently very popular approach that also can increase both the stability and the oxygen reduction reaction (ORR) activities of the catalysts [8-13]. In PEMFC, kinetics of hydrogen oxidation reaction, which is occur in anode catalyst layer (CL), is faster than the ORR in the cathode CL.
    2. Here isn’t the B in CB black?: In our previous work, CB black materials prepared by silica templet process showed promising properties as support materials in terms of both stabilities and activities for fuel cell applications [17].
  4. It would be useful if the authors state the reason for why the result in this statement is happening. Unfortunately, most of the proposed non-carbon based support materials are not transferable to the real fuel cell applications. Although the average Pt crystallite sizes of the homemade cat-130 alysts are larger than the commercial catalysts, they are still usable as fuel cell catalysts 131 since 2 to 5 nm Pt nanoparticles are usually used for fuel cell applications.
  5. I would explain this by stating that the particles can reduce their overall surface energy by agglomerating. Releasing their extra energy is an odd way to explain it because I don’t know what extra energy is. “Pt nanoparticles possess high surface energies, they have high tendency to release their extra energies by agglomeration when they stayed closer.”

Author Response

Dear Reviewer,

Thank you for reviewing the manuscript. I would like to answer your quistinaries in below.

Major comments

1. You state that at high currents that supports with low BET surface area are best. However the C65 support has the best performance when comparing all systems using the homemade Pt and it has a very high BET area. The Timcal has a lower BET surface area and has worse performance. I am confused about the conclusions made. I would strongly suggest to keep the thickness of the layers the same so it is easier to compare as well.

Reply: In the discussion part (293-300) it is now organized. Keeping same Catalyst layers thickness with same Pt loading (0.25mg/cm2) is not possible since different types of supports were used.   

2. The motivation for the work is also not apparent. Much of the introduction is spent on more general topics, but there is no discussion of the work that is similar to their own which there is a lot of. I would suggest to provide a thorough literature review of studies on catalyst supports and the effect of their surface areas etc. on performance. If this has never been done than state that and what is the work that is the most similar and what does it say.

Reply: We did not find any literature that compared the effects of BET and MEA performance.

Other comments

1. Figure 1 legend doesn’t correspond to one column in the table above it. Data on how the crystallite size was determined and the error would be useful.

Reply : corrected. Please see 126-127 and 329-330, A topas software with using Rietveld refinement technique was performed.  

2. Table 2 what is the high surface area CB? I don’t see it in table 1.

Reply : Table 1 showed all homemade supports, however table 2 showed supported homemade and commercial catalysts (TKE, a commercial catalysts with high surface area CB).

3. Figure 2 would be useful to explain why the one sample has particles that seem to decrease in size. Is this actually happening or is this a measurement error?

Reply : May be you wanted to mention table 2. Both TKE and TKV are commercial catalysts and possess very low Pt particle sizes (in our measurements and also found in the Data sheets). The highest ECSA is also an evidence of lowest Pt particles of TKE catalysts (see Figure 4).

4. Figure 3 why not normalize the current by surface area? This would be a more fair comparison. Is the quantity of current in this test really correlate with ORR kinetics? Why not measure ORR currents? Is this what you are trying to do in Figure with with MA? I am not sure what MA is, but you should show more about the KL procedure you used.

Reply : We followed the usual measurement procedure (ref 56). ORR activity is now changed to Mass activities (MA). MA measurement was done using KL procedure is discussed more in reference 47 and 56 (also mentioned in 351-356)

5. How is the optimum I/C found? Is that from literature or is that something you are measuring?

Reply : Figure S2 is an example of I/C optimization. Other catalysts were also done by the same way previously (204-205).   

6. Figure 6 is very sloppy. The figures should be the same size. You should be showing trends in fitted resistance and capacitance in addition to the Nyguist plots.

Reply : Unfortunately, we didn´t make fittings till now.  

7. Figure 7 should have the bottom removed and to only leave the scale bar. Then label on the images in white the sample and put them together on a figure that can be on one page.

Reply : There is a chance of deformation of scale. If needed for publication, we will do it.

8. Higher quality figures are expected for publication.

Reply : If asked for individual figures, we will do it.

9. You never discuss your supplementary figures in the main text.

Reply : Just relevant parts were showed.

Comments on writing

1. Acronyms in the abstract are not defined

Reply : now improved according to Academic Editor Notes

2. The first four paragraphs of the discussion section are not a discussion of your results

Reply : now improved according to Academic Editor Notes

3. The writing is understandable, but many sentences like the following couple I show here as an example need proofreading before being published:

    1. Introduction of surface functional groups on CB materials are recently very popular approach that also can increase both the stability and the oxygen reduction reaction (ORR) activities of the catalysts [8-13]. In PEMFC, kinetics of hydrogen oxidation reaction, which is occur in anode catalyst layer (CL), is faster than the ORR in the cathode CL.
    2. Here isn’t the B in CB black?: In our previous work, CB black materials prepared by silica templet process showed promising properties as support materials in terms of both stabilities and activities for fuel cell applications [17]

Reply : corrected

4. It would be useful if the authors state the reason for why the result in this statement is happening. Unfortunately, most of the proposed non-carbon based support materials are not transferable to the real fuel cell applications. Although the average Pt crystallite sizes of the homemade cat-130 alysts are larger than the commercial catalysts, they are still usable as fuel cell catalysts 131 since 2 to 5 nm Pt nanoparticles are usually used for fuel cell applications

5. I would explain this by stating that the particles can reduce their overall surface energy by agglomerating. Releasing their extra energy is an odd way to explain it because I don’t know what extra energy is. “Pt nanoparticles possess high surface energies, they have high tendency to release their extra energies by agglomeration when they stayed closer.”

Reply : corrected

Reviewer 2 Report

This study mainly handled the relationship between the specific surface area (SSA) of carbon supports and fuel cell performance. Unfortunately, throughout this work, I don't find out that the novelty of research findings, which already is well-known in this field. Also, I feel the lack of discussion on the presented data. 

So, I recommend that this work must be carefully revised, before submission to any other journal (including this journal). 

Author Response

Dear Reviewer,

Thank you for reviewing the manuscript.

I tried to improve the discussion part according to Academic Editor Notes.

Unfortunately, I didn´t find spcipic areas in your comments that I could improve.However, I tried to correct some areas in the reviewd manuscript.

Thanks

Author

Reviewer 3 Report

This article reported influences of surface area of carbon support material on performance of platinum catalyst. The report was strong and solid.

- The authors insisted carbon support with low surface area showed higher voltage at high current density. Various results were presented including polarization curves. But in order to confirm higher voltage at higher current density of PEMFCs, constant current measurement results of fuel cells are recommended.

Author Response

Dear Reviewer,

thank you for reviewing the manuscript. I have improved abstruct and discussion parts in the revised version according to Academic Editor Notes.

Thanks

Author

Round 2

Reviewer 1 Report

The authors have addressed many of the smaller concerns relevant to my review and I appreciate their efforts to address all of my comments. Unfortunately, my two major comments were not completely addressed and several of the smaller comments are not addressed.

1. You state that at high currents that supports with low BET surface area are best. However the C65 support has the best performance when comparing all systems using the homemade Pt and it has a very high BET area. The Timcal has a lower BET surface area and has worse performance. I am confused about the conclusions made. I would strongly suggest to keep the thickness of the layers the same so it is easier to compare as well.

Reply: In the discussion part (293-300) it is now organized. Keeping same Catalyst layers thickness with same Pt loading (0.25mg/cm2) is not possible since different types of supports were used.

Reply: It seems like the conclusions of your paper have changed now to medium support area being optimal, but it is still hard to understand how you can reach that conclusion with any certainty. The three support materials you are testing have BET of 63, 174, and 192. It seems to me that you really only have low (63) and high (174 and 192) surface areas and if the 174 has the highest performance than you should test a BET area around 120, which is not so close to the "high" and "low" BET samples.

The other major issue is that the conclusions are different at different currents and different powers, and those differences are believed to be due to Pt particle size. The lack of systematic testing of dependent variables with other parameters fixed makes understanding the results confusing and reduces confidence in the conclusions.

For example, when I read the updated abstract I have trouble understanding how you can reach the conclusions you state with certainty concerning the medium BET surface area having optimal porous structure, water management, and high Pt particle size.

At high current density, Pt supported on medium BET surface area supports showed a higher cell voltage than Pt supported on high and low BET surface area. Compared to the high BET surface area support, the high power density at high current loading was ascribed to maximum utilization of Pt, due to lack of porous structure and proper water management, while in the case of the low BET surface area support the better performance of the medium BET surface area support was due to the lower Pt particle size.

Before publication the tests and analysis need to be more systematically carried out and reported so that readers can easily understand what was done and the significant of the results.

2. The motivation for the work is also not apparent. Much of the introduction is spent on more general topics, but there is no discussion of the work that is similar to their own which there is a lot of. I would suggest to provide a thorough literature review of studies on catalyst supports and the effect of their surface areas etc. on performance. If this has never been done than state that and what is the work that is the most similar and what does it say.

Reply: We did not find any literature that compared the effects of BET and MEA performance.

Reply: Understanding of the effects of MEA parameters on fuel cell performance is an area that is well studied and should be thoroughly reviewed. During a quick review of literature a publication discussing surface area effects was located https://doi.org/10.1016/j.apenergy.2019.03.157

and there are many reports on MEA parameters that should be discussed to show the significant of this work.

https://doi.org/10.1016/j.apenergy.2019.113320

Smaller comments like #2 on Table 2 where both references to CB and Vulcan XC72 are presented, but it seems based on Table 1 that these are the same material. The labeling here is confusing.

Figure 7 still needs significant improvement as it goes onto several pages, the scale bars should be made easier to read, and labeling should be done on the image so a reader does not need to flip pages to find out what the image is of.

Overall, significant work is still needed for consideration for publication.

Author Response

Dear Reviewer,

Thank you once more for the comments. I would like to reply as below- 

Comment1 :

Reply : Yes, Medium BET supported catalysts are favorable for high current density, is the key point in this work. We mentioned medium that means, C65 and CB (BET 174 and 192m2/g) for homemade and TKV (250 m2/g) for commercial catalysts (see 286). High surface area supported catalyst is only TKE (800m2/g). we wanted to massage the reader that highest BET surface area support material is not feasible for fuel cell application at high current loads although it is widely using commercially.  We have chance to show here that C65 could be another choice of medium BET support as CB (which is thought to be only option for fuel cell catalyst manufacturer). GC is also a nice option for long time application when someone would like to operate the cell at low current loads (marine or stationery application where sufficient place is available for example).  

Comment2 :

Reply :

https://doi.org/10.1016/j.apenergy.2019.03.157 , Effect of MEA activation method in this literature (while using same materials) is discussed.  And  https://doi.org/10.1016/j.apenergy.2019.113320,  showed single cell and stack (4 cell) testing methods (while using same materials). However in our work, we kept the methodology same for all while using different carbon materials for comparison. Aim of this work is also clearly mentioned in 70-75, 268.   

Table 2 showed the list of homemade and reference catalysts. CB and TKV, both are Vulcan XC72 supported homemade and commercial catalysts respectively.

BET of of the CB (vulcan XC72) material we measured as 192 m2/g, however commercial data sheet showed 250m2/g (may be they did some oxidation before synthesis)

Figure 7  modified in one page

Reviewer 2 Report

This present paper can be published in this catalysts Journal. 

Author Response

Dear Reviewer,

Thank you once more for the comments.

I have updated slightly in this version.

Round 3

Reviewer 1 Report

I appreciate the authors attempting to improve the work, but my comments and significant concerns from the original review still have not been addressed thoroughly.